# Linking Flood Susceptibility Mapping and Governance in Mexico for Flood Mitigation: A Participatory Approach Model

**Rosanna Bonasia** [1,*] **and Simone Lucatello** [2]

[1] CONACYT-Sección de Estudios de Posgrado e Investigación, Instituto Politécnico Nacional, ESIA Zacatenco, 07320 Mexico City, Mexico
[2] Instituto de Investigaciones Dr. José Maria Luis Mora (CONACYT), 03730 Mexico City, Mexico
* Correspondence: rosannabonasia017@gmail.com

**Abstract:** In many countries of the world, floods continue to cause extensive damage to people and properties. This is also the case of Mexico, where meteorological phenomena cause flooding every year. In order to mitigate continuous losses and damages, crucial tools like hazard maps are essential for prevention. This review article analyzes the main reasons for the shortcomings on disasters caused by floods in Mexico. We argue that strong linking between the realm of technical hazard mappings and local governance as an integrated approach to manage disasters can be a basis for a new prevention policy in Mexico. This consideration is achieved through the description of the available information on the meteorological events that have caused major damage in recent years and the analysis of the interventions carried out at decision-making level by the government and the national civil protection system. The application that hazard maps have in the world and their usefulness is also discussed. From the analysis carried out it emerges that the inefficiency of the system in preventing damage caused by floods in Mexico is due to both the lack of high-level expertise in hazard maps design and the lack of their use in decision-making policies at the local level.

**Keywords:** flood risk; flood susceptibility mapping; governance; flood hazard; civil protection; disasters

## 1. Introduction

The recurrence of floods has increased worldwide, representing one of the most costly phenomena in terms of economic losses [1]. The increase in the hazard related to river flooding is due to several factors: population growth [2], climate change impacts as well as land change due to human influence [3]. Hoeppe [2], in his analysis on the trend of disasters related to natural events, first of all divides the various environmental perils into different groups (geophysical, meteorological, and climatological), and observes that, in 2014, the total number of events that encompass the three classes of hazards was of the order of 980. The author also states that the number of events that caused human or material losses grew from ~300 in 1989 to ~900 in 2014, with the greatest economic losses attributed to the storms (40%), followed by floods (25%). As a result of the climate change it is expected that the number of events of this type will continue to increase as the number of weather-related disasters.

The increase in world population has direct consequences in people settlements within areas prone to flood risks [4]. Urban floods are becoming a widespread problem, with a strong impact especially in terms of both direct and indirect economic losses. Many cities in developing countries are experiencing an unprecedented increase in population. This phenomenon is due to the mass movement from urban areas to the cities. This has led to the increase of human settlements, industrial areas, and infrastructures in hazard areas such as lands below rivers or inside dried up rivers. This is the case of

Mexico, where destructive river floods occur with annual recurrence, especially in zones occupied by large industrial cities with very high population concentration. Currently, floods in Mexico are a very delicate subject and difficult to treat, since they are the phenomenon that causes more economic damage in the country. Analysis of the National Institute of Statistics and Geography (INEGI) shows that ~41% of the national territory and 31 million of people are exposed to hydrometeorological phenomena. There are also various areas of the Mexican territory, far from the main urban centers, which are subject to flash flooding phenomena that occur particularly in the winter season due to intense rainfall that happen as consequence of cold fronts or convective storms and hurricanes [5,6]. Structural measures to protect against flooding are present in the Mexican territory, but are not uniformly distributed and almost completely lacking in areas of low economic interest. Furthermore, an education and prevention policy is absent or, in some cases, developing.

At the base of a fruitful prevention policy there must be the probabilistic study of the occurrence of the natural phenomenon, combined with the calculation of the damage it can cause, in the short- and long-term. This kind of study serves to provide the authorities with necessary elements to define possible risk scenarios and, in turn, to allow sustainable land use planning. A complete risk assessment includes the analysis of several factors like the hazard, aimed at the study of the nature of the phenomenon; the return period; and the exposure. Members of the international scientific community have, over the years, developed research solutions that improved the capabilities of flood propagation analysis models, with the determination of the characteristics and causes of the phenomenon with robust and reliable results. The short term prediction provides necessary information in the unrest period or during the emergency, since results of such modeling can provide a preliminary assessment with hours or days' advance notice. However, an appropriate risk management mainly requires a long-term assessment aimed to calculate the probability of occurrence of a given phenomenon and its impact, on a statistical basis. To this end, flood susceptibility maps are a fundamental tool to assess the expected impacts based on most probable scenarios.

One of the objectives of this review is to analyze the main natural, political and social causes of the recurrent occurrence of damage caused by flood phenomena in Mexico. This analysis is aimed at justifying the need for projects completely dedicated to prevention, focused on the use of flood maps that should be integrated within a participatory approach model between researchers, development workers, government agents, and local populations. In order to achieve these goals, the methodology followed in this work is as follows.

- The principal flood events that caused the greatest damage in the country are described. Given the great variety of hydrogeological context that Mexico presents, floods analyzed here are of various types: from pluvial to flash floods, also including the coastal and fluvial ones.
- Risk management and civil protection actions currently operating in Mexico are described, and the principal decentralized parts of the government system dedicated to emergency management are analyzed.
- Existing flood maps techniques and application in the world are reviewed.

On the basis of data acquired we perform an analysis of the impact of the use of flood maps in Mexico at social and administrative level and propose a hazard management model based on a participatory approach.

## 2. Diagnosis of Floods in Mexico

Mexico is subject to a great variety of flood phenomena, due to the vastness of its territory, and to the variety of hydrogeological contexts that characterize it. Pluvial floods are generated as a consequence of different hydrometeorological phenomena occurring within the Republic throughout the year. Because of its geographical position, Mexico is frequently subject to extreme weather phenomena that produce severe convective storms, tropical cyclones and cold fronts. The country is located within a band of low pressure that forms on the regions of masses of warm water in the

tropics known as the Intertropical Convergence Zone, which is characterized by the formation of intense rains that produce floods year after year. Mexico is also affected by river floods that verify when the water that overflows from rivers remains on the surface of lands around them. When this happens, the volume of water that drains on the land through the channels increases with the area of the basin. This occurs mainly in the rivers with major length and in those that reach the coastal plains. Coastal floods are also very common, due to the presence of tropical cyclones that generate storm surges. Although less frequent, Mexico has also been affected by floods caused by hydraulic infrastructure failures.

During the period between 2000 and 2016, in the Pacific Ocean, on average, 17 tropical cyclones formed, causing floods and landslides in many coasts of the country. In the Atlantic Ocean, the average of the cyclones that formed in the same period is ~15, giving rise to similar consequences in terms of damage to the coast.

Floods in Mexico are not just a contemporary problem. Arreguín-Cortés and Cervantes-Jaimes [7] collected historical information about flood events in the country and report that the first major flood that hit Tenochtitlan (ancient Mexico City) occurred in 1446. One of the consequences of the conquest was the supremacy of a European-style city conception that originally developed in an arid and dry space. The blinding of channels and the deforestation of the Valley of Mexico caused a natural imbalance that was expressed with the occurrence of constant floods that affected the city. Table 1 shows the historical most destructive floods occurred in Mexico.

**Table 1.** Historically documented major floods occurred in Mexico (modified after Arreguín-Cortés and Cervantes-Jaimes [7]).

| Year of Occurrence | Flood event |
| --- | --- |
| 1446 | The great flood of Mexico-Tenochtitlán caused mortality and severe damage to the chinampas [1] system in Mexico City, during the reign of Moctezuma I. After the flood there were shortages and hunger. The situation did not normalize until the 1455. |
| 1555 | The first great flood of colonial Mexico; as a consequence of which a hydraulic structure was built following indigenous techniques. This work provided some help, although it was not enough to completely solve the problem of floods. |
| 1629 | This event started on September 21 and was caused by a heavy rain that lasted approximated 40 hours and resulted in the total flooding of the city. The water level reached over two meters and the city remained flooded for five years. |
| 1760 | This event is known as the "Guanajuato disaster". The heavy rains that fell for three hours caused the overflowing of the Silao river. |

[1] A chinampa is an ancient Mesoamerican method of agriculture and territorial expansion that served to cultivate flowers and vegetables.

Apart from the geographical location of the country, an element that increases the risk of flooding in Mexico today is the presence of dams. Throughout the Mexican territory there are approximately 800 large dams and more than 4000 small dams, most of which currently represent a hydrological risk due to various factors, including the clogging of pourers, deterioration of the curtain due to aging and lack of maintenance, leaks in the curtains, among others. Most of the dams are currently at the end of their designed life so that their overflowing during extreme rainfall can generate a higher impact downstream due to the combined effect of the dam overflow and the excess rain.

Based on the analysis of the trajectory and frequency of the hurricanes, as well as rain patterns and the risk of dam overflowing, the areas with the highest risk of flooding in Mexico were defined (Figure 1).

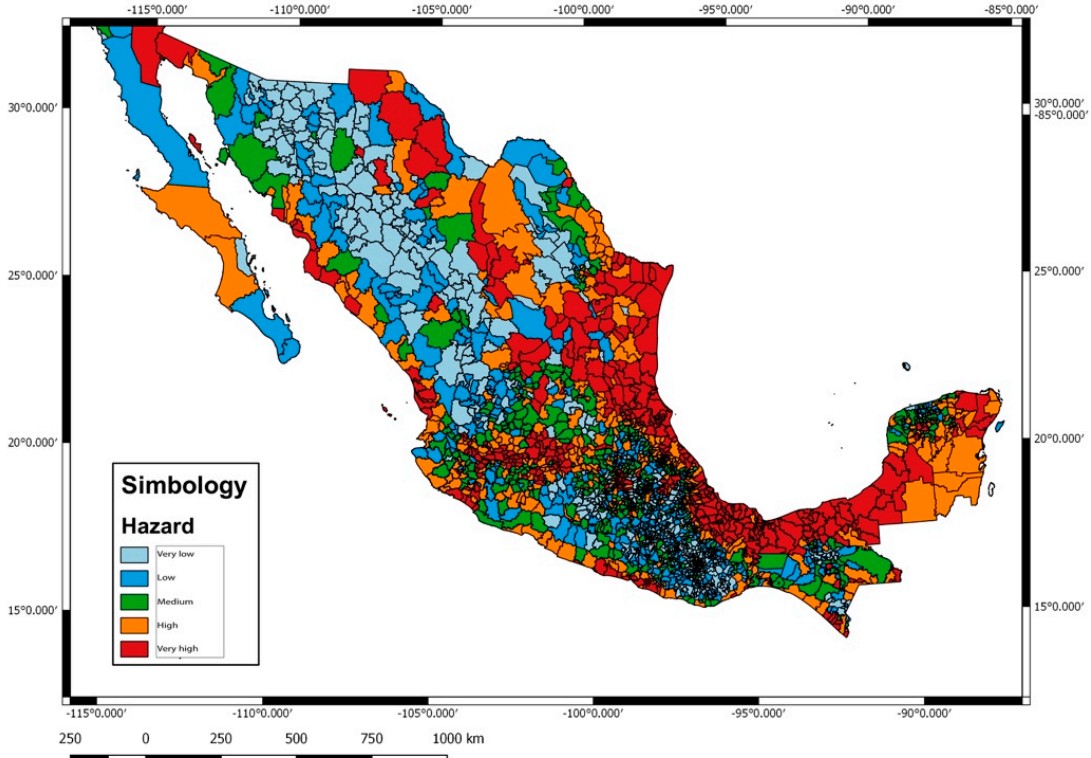

**Figure 1.** Areas of flood hazard in Mexico. The map is constructed on data of the Institutional Program of the National Institute of Educational Physical Infrastructure 2014–2018.

More than 8% of the national territory is subject to high risk of flooding. This implies a very high socioeconomic impact considering the effect that floods can have on urban areas, population density, economic activity, and existing infrastructures. According to the Executive Summary of Socioeconomic Impact of Disasters in Mexico during 2015, developed by the Ministry of Interior [8], the damage caused by hydrometeorological phenomena represents 60% of the total damage caused by natural events in the country.

Floods can be the consequence of slow or rapid processes. Some basins are characterized by having a slow hydrological response. This commonly occurs in areas where the slope of the channel is small and, therefore, the capacity of the rivers decreases considerably causing overflows that produce floods in the surrounding areas. In these cases, floods are generated in a relatively long time (several hours or days). On the other hand, sudden and intense rains can cause small currents that can transform, in a matter of minutes, into violent torrents capable of causing great damage. This phenomenon, known as flash flood, happens mainly in urban areas where the presence of buildings prevents the infiltration of water and practically all the precipitated volume becomes runoff. This flood can also occur in basins located near steep mountains where valleys, ravines and alluvial fans exist. Moreover, flash floods can be the consequence of dams breaking.

An example of slow processes verified in the country, was the flood that took place in the state of Tabasco and affected mainly the capital city, Villahermosa, in 2007 (Figure 2).

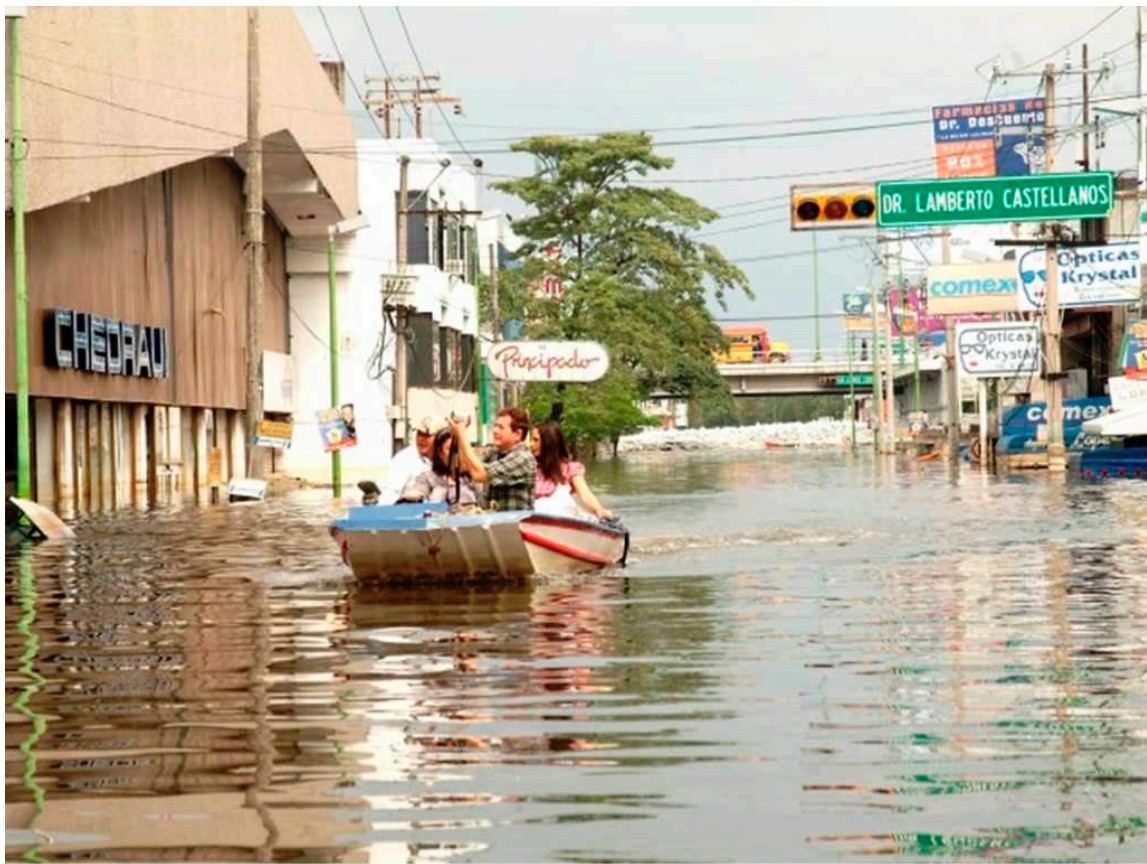

**Figure 2.** The city of Villahermosa flooded for 80% of its territory on October 28, 2007 (Tabascohoy).

Between October 28 and 30, 2007, intense and extraordinary rains occurred in the Grijalva river catchment. Rains reached depths greater than 50 mm in some locations of the basin of the Peñitas dam, causing the need to open the dam floodgates. Other rivers of the drainage system that flows into Villahermosa also reached high levels, close to the critical thresholds defined for issuing alerts [9]. Generated floods covered an area equivalent to 70% of the Tabasco state territory affecting more than 1 million people. An example of rapid (flash) flooding was the event that occurred in October 2008 in the south of the Sonora state, as consequence of the hurricane Norbert. The amount of water precipitated in the mountain range caused runoff from the peaks of the mountains downstream to the coastal plains. The discharge energy of the runoff was so intense that materially destroyed vegetation, soil and rocks along the flow path. Heavy rainfall in 2010 generated by the passage of tropical depressions caused severe flooding in the states of Tabasco, Oaxaca, Veracruz, Zacatecas, and Chihuahua, affecting more than 10,000 people and many sectors of federal, state and municipal highways. Hurricane Jova, in 2011, affected the states of Colima and Jalisco (Figure 3), leaving communities isolated by the overflow of rivers; roads were closed and thousands of houses in coastal and mountainous communities remained without power and drinking water.

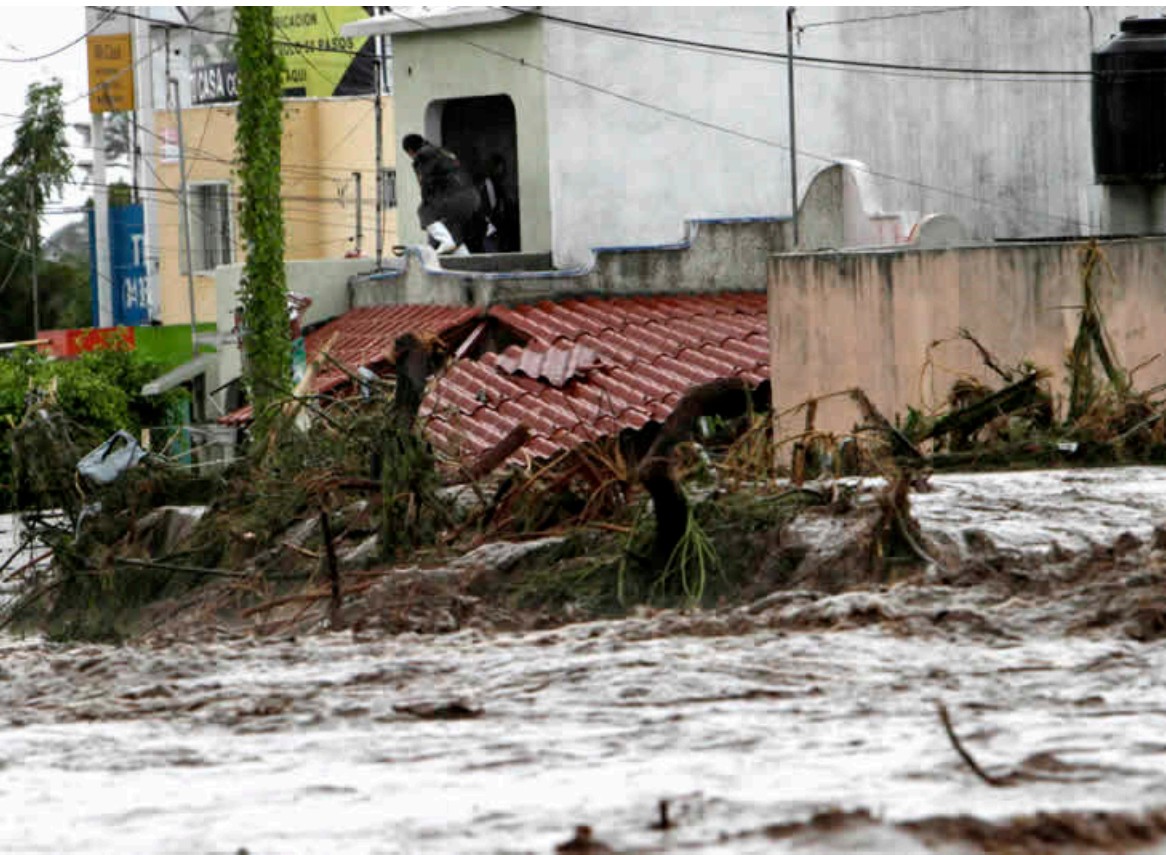

**Figure 3.** Damage caused by hurricane Jova in the Jalisco state in 2011 (El Universal).

In 2013, the incidence of two simultaneous tropical cyclones in the state of Guerrero (Ingrid in the Gulf of Mexico and Manuel in the Pacific Ocean) seriously affected the mountain region, which presented greater losses due to its historical conditions of vulnerability, environmental deterioration and the precarious conditions of the houses. Heavy rains in October 2015 caused damage in the state of Veracruz and generated floods in Chihuahua and Tamaulipas.

The Mexican territory presents a huge variety of hydrogeological contexts, so the causes of floods have to be found in the specific characteristics of each zone. Every year, large rivers in Mexico cause floods as a consequence of heavy rainfall, but also because of the loss of the hydraulic capacity of the channels. In recent decades, deforestation in Mexico has grown at an alarming rate [10]. According to the Institute of Geography of the National Autonomous University of Mexico (UNAM), each year Mexico loses 500 thousand hectares of forest and jungle. Porter-Bolland et al. [11] states that, in the period between 1977 and 1992, the deforestation rate of the Tropical Dry Forest reached 1.9% per year. As an example of intense change in land cover, between 1970 and 2010, the Acapulco Bay in the state of Guerrero lost approximately 110 km$^2$ of natural vegetation, increasing the flood risk from low to very high in more than 40% of the area [12]. The deforestation of the Guerrero mountainous region doubled the number of floods, affecting principally Acapulco, the capital city [12].

The ineffective urban management also increased the number of floods in the country. Uncontrolled urban development magnified the hydrologic land response to precipitation, increasing flash floods. One example is Villahermosa, the capital city of the Tabasco state. Until the 70s, Villahermosa was settled on a hill between two rivers: Carrizal and de la Sierra. In the last 50 years, the urban zone increased in an uncontrolled manner along rivers margins, extending within the areas more prone to floods [9]. Although levees have been constructed to protect the city, they are not sufficient. Areu-Rangel et al. [13] recently proposed three urban expansion scenarios for 2050 in Villahermosa, on the basis of the estimations of the city's growth rate and numerically forecasted inundation levels.

Their results show that a future urban expansion would cause inundation depths to rise up to 0.7 m in areas that are already affected by floods every year. The urbanization does not only concern the large Mexican cities, but also the areas surrounding the metropolitan centers. These areas, defined "peri-urban" or "rur-urban" [14], are constituted by small towns and little cities that distribute around a metropolitan core. This is what happens for example in the Upper Lerma River Valley in the state of Mexico. Here the precipitation has increased in the last decades [15], but more importantly, the socioeconomic conditions have increased the risk of flooding [16].

In the desert and semi-desert areas, less intense rainfall makes floods less frequent. However, extraordinary precipitation can cause serious damage. Large flows are caused by tropical cyclones that generally occur in the month of September. This was the case of the heavy rains that struck, in September 2018, the city of Torreón in the state of Coahuila (Figure 4). The metropolitan zone is located in a desert area with little rainfall per year. The National Water Commission indicated that, in September 2018, 270 mm of water fell, in contrast with the 18 mm that usually fall in this month.

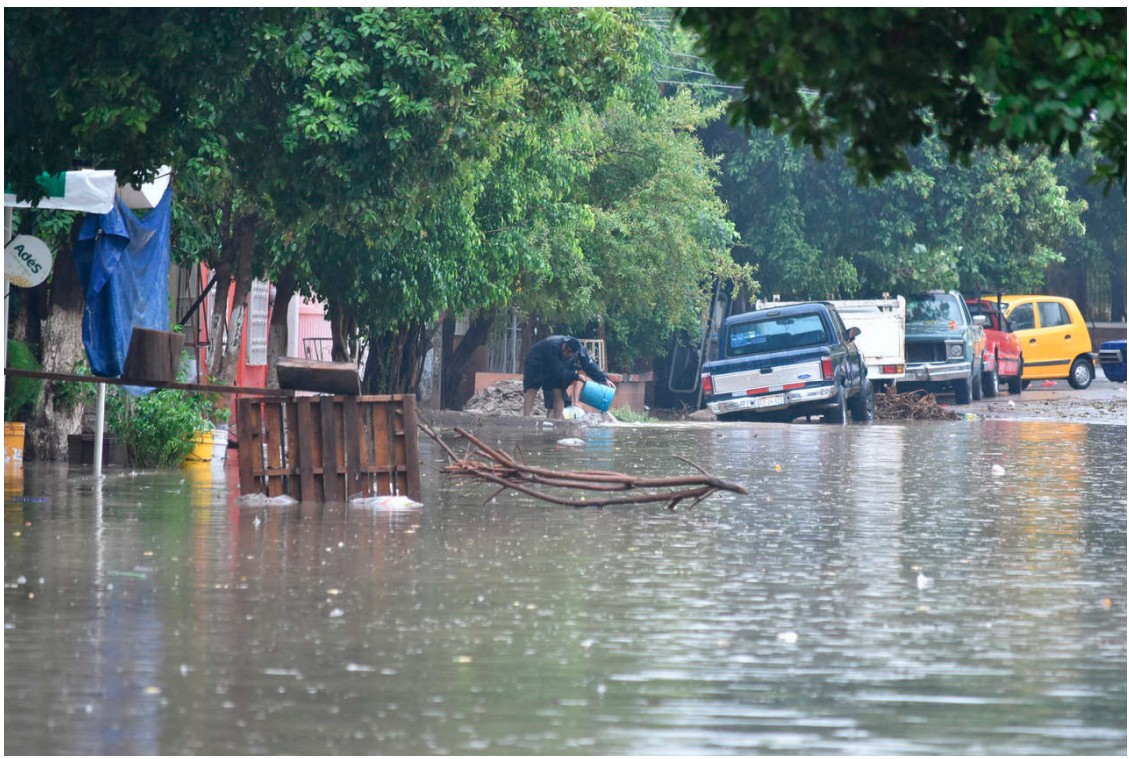

**Figure 4.** Floods in Torreón, 28th of September 2018 (El Siglo de Torreón).

## 3. Risk Management and Existing Civil Protection Actions in Mexico

On May 6th, 1986, the National Civil Protection System (SINAPROC by its initials in Spanish) was created in Mexico in response to the need to deal with disastrous events like the catastrophic earthquakes occurred on September 1985. The SINAPROC is a figure of multi-institutional coordination, in which the government, together with the civil society, organizes the governmental functions in terms of civil protection in order to protect life, the environment and the heritage of society. In case of an emergency due to the approach of an extreme meteorological phenomenon capable of generating floods, the National Water Commission (CONAGUA), through the Unit of the National Weather Service (USMN), establishes a permanent surveillance system, which provides bulletins and predictions to the SINAPROC. The National Center for Disaster Prevention (CENAPRED) also draws up bulletins based on the rain forecast of the USMN, to inform about the possible risk of flooding in the municipalities of the country. The National System of Civil Protection is coordinated by the Government Secretariat

(SEGOB), which is the agency responsible for directing the mechanisms and policies for prevention and attention to risk, disasters and consequent crisis.

The sum of regulations, programs, and policy actions created over the past three decades were put under pressure once again in September 2017, when new earthquakes hit the Mexican territory and Mexico City. For its scattered nature, the SINAPROC was not able to respond properly to the threats posed by the earthquakes emergency. The mandate for coordination between multiple governmental organizations at national, provincial and municipal levels, and several specialized federal agencies (such as the National Center for Disaster Prevention, Mexican Petroleum, and the National Water Commission, among others) are currently under severe transformation for the limited response during the 2017 emergencies. In particular, the SINAPROC's top-bottom government structure, based at the Ministry of Internal Affairs (SEGOB), is in charge of dealing with specific emergency management actions and the implementation of different programs through coordination with provincial and municipal levels of government [17,18]. Thus, SINAPROC is an organized group of structures made of functional relationships, methods, and procedures involving all levels of government that also engages the private sector and non-governmental and civil society organizations [19]. It is therefore evident that Mexico's natural hazard management and risk reduction responsibilities are divided among multiple agencies as clearly stated in the General Law of Civil Protection [20]. In addition, Mexico's government structure is formally decentralized and provincial and municipal civil protection agencies are supposed to be strategic components of SINAPROC. However, municipal governments have very limited capabilities in terms of involvement, level of participation and effectiveness as well as expenditure on civil protection, urban planning and public works in developing coherent and robust local civil protection actions. This results in very weak hazard detection systems and insufficient training and knowledge of natural hazards. As stated by Saldaña-Zorrilla [21] and Estrada [22], the civil protection system has made little progress over time regarding emergency management tools, like flood hazard maps. Problems for integrating strategic responses to disasters are clearly dispersed in different governmental areas and this may have direct consequences on coordination mechanisms and responses at local level. As long as the way in which the reactive approach to risks management remains focused on reaction rather than prevention, all civil protection actions will keep undermining the domain of risk control and hazards detection [23].

For the purpose of this work, it is worth remembering that, parallel to SINAPROC, there is, in Mexico, another institutional organization, the Climate Change National System (SINACC), which is more recent than the Disaster Risk Reduction (DRR) framework: Disaster risk reduction is a framework concept and practice used for reducing disaster risks through systematic efforts that understand and curb the causal factors of disasters. Reducing exposure to hazards, lessening vulnerability of people and property, wise management of land and the environment, and improving preparedness and early warning for adverse events are examples of the disaster risk reduction framework (UNDRR), upon which the Mexican Civil protection system is based. An Inter-Ministerial Commission for Climate Change (ICCC) was created with the effective entry into force of the Kyoto Protocol in Mexico in 2005 [24], and it coordinates several actions within the National System for Climate Change (SINACC). Moreover, the National Institute of Ecology and Climate Change (INECC), a decentralized agency that was recently created following the promulgation of the National Climate Change Law in 2012, mainly focused on applied climate and environmental research, head the SINACC. Within its mandate, SINACC plans and collaborates in the development of strategies and measures related to national Climate Change (CC) goals such as the National Strategy for Climate Change and the Special Climate Change Program; it also makes recommendations on policies and actions of mitigation or adaptation to CC for provincial and municipal governments. Finally, it recently launched the first national vulnerability atlas which is an instrument to measure vulnerability at hazards related to climate change in Mexico. It is worth noting that, despite the similarities between SINAPROC and SINACC, the latter has been much more focused at the national scale and on the implementation of CC targets in the different economic and social sectors represented in the national ministries, such as

energy, agriculture, infrastructure, and urban development. Currently, a new law that may integrate SINAPROC and SINACC implementation schemes is under discussion at the Mexican Congress. It will guide policy-making over the next decades. As Aragon-Durand [25] points out, despite the similarities between the two systems, the decision-making flow within each of them is different. In the case of SINAPROC, it was designed hierarchically, compared to the decentralized scheme of SINACC, in which the different ministries that participate in the system and their provincial offices influence local climatic policies' decision-making and implementation without coordination with municipal governments or other local initiatives. In addition to these mandates, there is currently a program to generate Municipal Climatic Action Plans (PACMUN), which have been elaborated and delivered to local governments through a partnership between INECC and local non-governmental organizations (NGOs) [26].

As highlighted in the last Intergovernmental Panel for Climate Change (IPCC) Assessment Report, countries have been encouraged to address the shortcomings of such a "silo approach". An integration system that merges both CC and DRR is currently under discussion in Mexico, and it will be probably adopted under the new Civil Protection Law, due in the next two years. This may constitute a step forward in terms of national development planning by inducing all federal governmental levels to combine the two approaches into a new national and renewed framework for risk management [27].

It is also worth reminding for the purposes of this analysis, that Mexico's government structure is decentralized. It means that beside the federal structure level, the provincial and municipal system have civil protection agencies that are key components of SINAPROC. According to the national law on civil protection, municipal governments hold the responsibility to first respond to an event through the local civil protection office, and this level of government is also in charge of public services delivery, some social development programs, as well as all urban land use provisions and many public works [26]. Therefore, municipal governments represent the first government structures that respond to natural hazards like earthquakes, volcanic eruptions and flooding. However, municipal involvement, effectiveness, and level of participation of population in building resilience to flooding is asymmetrical; for instance, when facing severe storms and sudden flooding, municipalities' capacities and civil protection reactions look very weak and disarticulated. This is due, among other factors, to the lack of institutional coordination between levels of governments, lack of resources, poor prevention systems, and very feeble capacity building to use tools like hazard maps provided by the scientific community.

Concerning floods, the 2013–2018 National Development Plan addresses the two issues of CC and DRR separately although it seems to encourage integration. Examples of these incipient efforts for policy integration are the creation of an Inter-Ministerial Committee for Drought and Flood events (CIASI), which coordinates actions amongst Federal Government entities regarding risk analysis and the implementation of prevention and mitigation of extraordinary meteorological phenomena and their effects, such as drought and floods [24]. Additionally, in the last few years there has been an inclusion of CC officers and planners from INECC into the social and natural science scientific committees for public policies and planning at CENAPRED; in this regard, current discussions on risk reduction policies at CENAPRED are increasingly framed and planned with CC mitigation and adaptation actions in mind. As can be seen from this brief description, the interplay between the DRR and CC mandates and institutions is essential for the future of both policy fields, but is still disperse in terms of scales and scope.

In this complex context of actions aimed at mitigating the risk of natural phenomena including flooding in Mexico, the attention is called to the urgency of realizing and using hazard and risk maps. Hazard maps should not be limited to a synchronic vision of the risk; they should constitute an obligatory reference for the disaster management as well as the assignment of resources. According to Medina Barrios et al. [28], currently, only 15% of the municipality of the country has an atlas of risk that complies with the regulations of CENAPRED or the Secretary of Agricultural, Territorial and Urban Development (SEDATU). As part of the flood control plans, Mexican legislation, through the operation of the Natural Disaster Fund (FONDEN), provides resources for the sites impacted by

hydrometeorological phenomena, which are meant to allow the reconstruction of housing, roads, and hydraulic infrastructure in order to return to normality after a disaster.

Following the damage caused by hurricane Pauline in 1997 in the city of Acapulco, Guerrero, CENAPRED developed the project of a Hydrometeorological Warning System (SAH), which is a telemetric network consisting of rain gauges distributed in the basin. It aims to monitor the evolution of rainfall and water levels in the rivers. If critical alert thresholds are reached, the system issues warnings for civil protection within few hours. Systems similar to the one designed for Acapulco have been created later and are currently operating in 13 cities of the country, with automatic rainfall and hydraulic stations; for example, in Tijuana, Monterrey, and Villahermosa, to name a few [29]. The Federal Commission of Electricity (CFE) uses similar networks to monitor hydroelectric dams and mitigate infrastructure damage in case of extreme contingencies. The efficiency of the SAHs depends mainly on the status of the rain gauges (which present defects in many zones) as well as on the availability of precise hydrological models of the basins. Furthermore, applying this system to all the rivers in the country that are prone to overflowing implies high costs as well as an efficient collaboration and homologation of tasks among the bodies in charge such as CONAGUA, CFE, CENAPRED, and SMN [18].

To carry out actions against the damages caused by floods, the Federal Government introduced structural measures into its operational programs. Strategic infrastructures have been constructed in the Mexican territory to mitigate the negative effects of floods. The most widespread works for the control of floods in Mexico are the perimeter boards, which are compacted clay embankments, built with the purpose of protecting houses, agricultural land, and human lives against the action of water. These boundaries partially or completely surround the urban centers and have the advantage of being the cheapest solution; in addition, they do not alter the levels of runoff. Other infrastructures present in the country are the longitudinal boards, which are built along the banks of the river and serve to simultaneously protect several cities and towns, as well as large tracts of land with high agricultural and livestock production. During the floods, longitudinal edges completely modify the runoff in the territories both upstream and downstream. Permanent deviations have also been constructed to divert water from a river and lead it towards the sea, a lagoon or other channels. As mentioned above, dams are widespread in the country. They are designed to store excess water during the rainy season, which will be used during the dry season so to guarantee irrigation, potable water supply or the generation of electricity throughout the year. Dams have been also introduced with the sole purpose of controlling floods. However, they resulted to be responsible for increasing the likelihood of flooding. According to an evaluation of the state of dams in Mexico, described in the National Inventory of Dams [30], the main anomalies observed in Mexican dams are the obstruction of pours, causing overflow through the curtain and its possible rupture; deterioration of the curtain due to aging and lack of maintenance; anisotropic flow, occurring when the horizontal and vertical permeability are different; and filtration in the curtain. More than 3% of dams in Mexico can currently favor floods [30].

As clearly stated in this section, risk quantification and mapping is necessary for developing risk management actions at different scale and governance levels. Vulnerability to intense rainfall in Mexico has become a constant over time. The union of the effect of constant change in land use and human interventions, with no risk prevention planning, is leading to an increase of disasters. As shown by Magaña at al. [31] research results indicate that the probability of floods grows as risk increases. In Monterrey for instance, the hazard activity has not increased in recent decades, but risk has risen due to the increased vulnerability. Another clear example of the risk-modulating effect is Boca del Río (Veracruz), where even when the hazard activity remains almost constant, the modulating effect of vulnerability increases the flood risk value, just as the flood activity increases [32]. In order to reduce these risks, it is crucial to better define hazards mapping which may contribute to understanding increase vulnerability to intense rainfalls as well as working with the evolution rate of the vulnerability factors in the given area.

A community-based flood hazard mapping technique can help to address the current gap solving the defects of the top-down approach in disaster planning as well as fostering all stakeholders' participation in an integrated development of the mapping process [33].

## 4. Flood Maps and Existing Applications

Hazard maps play a decisive role in flood prevention plans, given their strong space-time component [34]. An effective assessment of flood risk involves the analysis of floods in terms of area and time. Analysis of historic records and flood data have been widely used to estimate flood hazards, especially in cases when hydrologic models that use time series flood data to calculate flood depths cannot be applied. The analysis of topographic and geomorphological data variation in areas subject to flooding provides indications regarding the spatial distribution of the phenomenon that, combined with the temporal variation, represents an important tool for the flood hazard assessment. Risk maps help in long range forecasts as well as in the urban planning of rapidly growing urban areas [35]. Following the definition by Kron [32], the risk should be considered as the product of a hazard (understood as the occurrence probability of an event) and its consequences. On this basis, a risk-based approach is the only instrument able to obtain and analyze flood frequency and magnitude as well as its consequences. Flood maps can be created with the aim of separately analyzing risk or hazard. Hazard maps focus on the calculation and display of the probability of occurrence of an event and its magnitude. Risk maps also contain information on the damage that the given event can cause. Both these tools are fundamental in the management of the flooding. The construction of hazard maps is based on the estimate, through statistical models or hydrological methods, of discharge rates in correspondence of specific return periods. Flood levels are calculated by means of 1-D or 2-D hydrodynamic models. Finally, flooded areas are obtained by coupling water levels with digital elevation models (DEMs). The combination of the information related to the hazard with the direct consequences of it, produces risk maps, in which the economic damage is quantitatively calculated and indicated. The methods currently in use for the construction of flood maps have reached acceptable levels of accuracy. Epistemic uncertainties have been reduced improving numerical techniques, while random errors in field measurements have been overcome by introducing detailed statistical studies of historical rainfall and flood data, as well as studies on the effects of DEM properties (vertical and horizontal resolution) on flood models. In many cases, satellite images, when available, drastically reduce the error due to uncertainties of topographic data. It is important to mention that, recently, new techniques have been developed to make risk maps more precise. Data-driven models are gaining much importance along with physical and numerical models. For example, Machine Learning models are data-driven techniques that can be used for prediction purposes, with minimal inputs, on the basis of historical data. This technique has been applied for short and long term prediction and represents a forecasting method with a high level of accuracy [36]. Recently, several models have been developed for calculating flood susceptibility. Choubin et al. [37], for example, employ the multivariate discriminant analysis and the classification regression trees within the support vector machine, as a new method to create flood susceptibility maps. Novel hybrid neural networks models based on ensemble empirical mode decomposition and discrete wavelet transform have been developed to predict rainflow [38]. Advances in simulation of daily rainfall–runoff have been made using hybrid machine learning models such as least squares support vector regression and extreme learning machine [39].

Since the end of the 1990s, separate governmental organizations all over the world have financed projects to create flood maps. In contrast to traditional strategies aimed at protecting areas hit by recurring floods, some European countries recognized the need to address flood management on a risk-based approach. The already undertaken approaches rely on the effectiveness of studying the probability of occurrence of a flood in order to reduce it and consequently reduce the damage that may cause. de Moel et al. [40], provide a detailed overview of the flood management projects undertaken in Europe and existing flood maps. After the severe flooding that struck Europe in the period between 1998 and 2007, which caused more than 50 billion economic losses and more than 1000 deaths [41],

the European Directive on the Assessment and Management of Flood Risk was created [42]. According to this Directive, in order to reduce and manage the hazard posed by floods to human health and economic activity, after an accurate study of the river basins subject to flood risk, the creation of hazard maps is essential to establish flood risk management plans focused on prevention. From the creation of the directive to date, almost all European countries have flood maps, and, in some cases, guidelines have been established to create a unique methodology that avoids differences in maps formats.

In 2002, the Federal Emergency Management Agency (FEMA) of United States implemented the National Flood Insurance Program (NFIP) [43] aimed to purchase insurance as a protection against flood losses in exchange for States floodplain management regulation. In this document, the importance of flood hazard maps is underlined by the need to provide data necessary for floodplain management programs and to rate new construction for flood insurance. The National Flood Damage Reduction Program [44] started in Canada in 1975 to "discourage future flood vulnerable development" and allowed the creation of flood maps for more than 900 communities.

In all the programs mentioned, flood maps were used by governmental institutions in different ways; e.g., to inform the public and decision-making bodies and to provide guidance in the implementation of rescue and reconstruction operations. Emergency plans, for instance, have found a concrete basis in the use of hazard and risk maps. Jonkman [45], in his overview on risk measures to reduce loss of lives and economic damage, underlines the importance of the analysis of flood patterns, together with the probability of inundations as a key for damage models and to estimate the potential loss of lives. In Norway and Sweden, as well as in Finland and U.K., flood hazard maps and risk information have been used for spatial planning where binding legislation exists and as an informative tool for decision-makers.

Flood maps are also used as an instrument to educate and create knowledge about the phenomenon and its consequences [46]. Several countries in Europe and US have developed awareness and information campaigns, in which online flood maps have been created, and are available to be consulted by people. This type of tool is essential for the population living in flood-prone area, since it describes the hazard and its impact and consequently provides the elements to protect themselves and deal responsibly with the risk. The NFIP for example, produced the FEMA Flood Map Service Center (MSC) [47], a public source to obtain flood hazard information. The MSC can be used to access official flood maps and different flood hazard products, but more importantly, it is a tool that provides explanations for understanding the risk of flooding. FEMA also provides a methodology for estimating potential losses from disasters generated by earthquakes, floods, hurricanes and tsunamis [48]. The Irish National Flood Hazard Mapping Website [49] provides access to flood plans and flood maps developed by the Irish Office of Public Works, and gives information on flood risk management in Ireland. An example on how to use the platform is shown in Figure 5: the user has the possibility to consult the measures for reducing flood risk for specific location (structural flood relief works and works to improve channel conveyance or the storage and diversion of flood flows).

In Mexico, the first extensive flood and coastal risk study and subsequent adaptation assessment was recently carried out by Haer et al. [50]. In this work, the authors apply a flood risk analysis in Mexico, using a global dataset coupled with a cost–benefit analysis. The authors highlight how the application of a methodology that allows the estimation of the future hazard and risk characteristics, would bring economic advantages in flood protection strategies. The results of this work show that, under current climatic conditions, the risk of river flooding is higher than the coastal one and it is estimated to increase to 2 billion USD year$^{-1}$ in 2080. The importance of the study realized by Haer et al. [50] is evident when considering that it provides a detailed estimate of how to increase, for example, the protection systems and consequently decrease the economic damage caused by the recurrent flooding in Mexico. Recently, studies have also been carried out to analyze the main causes of floods in the areas at great risk in Mexico and implement approaches to the creation of hazard maps. In any cases these studies are still few and fragmented: they do not provide a unique methodology that can be applied at the level of the whole country. For example, Norman et al. [6] assessed the

flood vulnerability of Nogales, Arizona and Nogales, Sonora, in Mexico, implementing a model to evaluate watersheds. Hernández-Uribe et al. [51] applied a methodology for flood risk analysis in the urban basin Atemajac, in the Jalisco state. As a result of this work, zones under flood risk and high vulnerability have been delimited. Zúñiga and Magaña [12] studied the effect of land use cover change on the flooding risk in Mexico. They compared observed and modeled flood frequencies for the period 1970–2010, and found that, recently, the most frequent floods are the result of a rapid and intense deforestation process. Bonasia et al. [52] realized a flooding hazard assessment at Tulancingo in the state of Hidalgo. The authors developed a methodology for evaluating downstream flooding probability and associated inundation maps to support long-term mapping and minimize flood impact. Their study is based on a key factor that increases the risk of flooding in Tulancingo which is represented by the presence of a dam located at few kilometers from the municipality. Results show that low levels of impact are unlikely to occur, while high flow rates have a high probability of occurrence. Finally, Areu-Rangel et al. [13] assessed the impact of change in land use and urban growth on the flood hazard in Villahermosa, Tabasco. They calculated flood discharges for different return periods considering land uses of the catchments as they were in 1992 and as they are today, and estimate, by means of a 2D shallow water model, the increase of water depths in the city from 1992 to the present. Results show that the increase in flood depths due to changes in land use in the basins reached 22%.

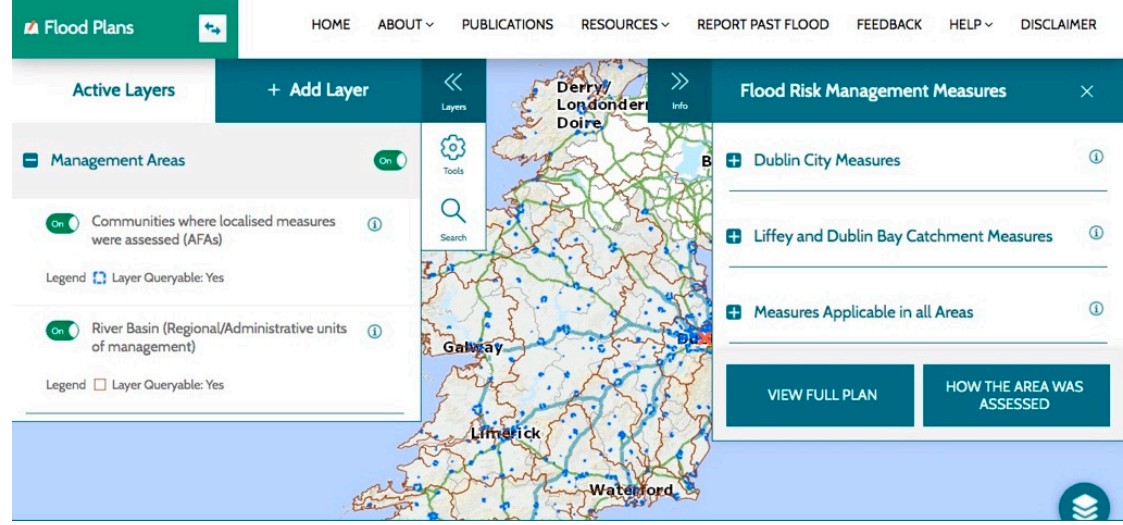

**Figure 5.** Example of the Flood Plans tool provided by the Irish National Flood Mapping Website.

In 2014, CONAGUA established guidelines for the preparation of flood hazard maps as a basis for the assessment of flood damage costs [53]. These guidelines define that the basic information for the construction of flood hazard maps should include topographic information, hydrological and hydraulic study, and damage analysis. The guidelines served as a basis for processing the National Risk Atlas, carried out by CONAGUA, which are published in the platform of the National Atlas of Risk (ANR) of CENAPRED [54]. The site only provides vulnerability index and flood hazard maps. However, as previously mentioned, not all regions of the country have complete information to generate risk maps.

## 5. The Impact of Susceptibility Flooding Maps in Mexico at a Social and Administrative Level: Participatory Approaches

As shown above, there is a plethora of risk flooding tools and maps available. However, the link with the social dimensions of risk is an unfinished task. The development of mapping floods exercises is complex and multidisciplinary and it includes technical expertise as well as human and financial resources. Currently, there are very few works and publications available that provide

guidance to develop a program on flood mapping and flood risk assessment with a social dimension included [55]. In the case of Mexico, flood maps can play an important role in decision-making, planning and implementing responses to flood risks. Community participation in capacity building to prevent flooding has been, for many decades, at the core of any international, national, and local development policy; emergency intervention involving people in many parts of the world, based on the assumption that a "top-down" approach is not adequate for its implementation, is also well understood. In this regard, the use of preventive tools such as flooding maps in Mexico should be coupled with a "bottom-up" approach, meaning that it should not be imposed by authorities but instead built from grassroots through assisted development strategies among stakeholders. This can be considered as an important step towards enabling local and vulnerable communities to be recognized as active actors in building and understanding local mappings with the help of scientific-expertise with the aim to foster participatory approaches in vulnerable communities [56]. It is a process whereby the communities concerned function and contribute to perform a predetermined activity as a cohesive group, while recognizing and enhancing the differences within them. This paradigm is also known as "Participatory Approach" [57]. Participatory approaches are a product of long-lasting interaction between researchers, development workers, government agents, and local populations [58]. Participatory methods at local-level began in the late 1970s with the introduction of traditional approaches like the Rapid Rural Appraisal (RRA), which was adopted in many poor areas and driven by processes of international cooperation and development with local decision-makers. This strategy recognizes the close collaboration with local populations, by collecting first-hand data from the local people about their perceptions of the local environment and living conditions in rural areas. Teams of researchers and local community work together with specialists and decision-makers to specifically build scenarios for responding to different threats. An important part of this methodology requires better communication processes with persons holding different degrees of knowledge and literacy, as well as visualization tools like simple symbols, or more sophisticated instruments like mappings. However, in critical terms, until the 1980s, such approaches had severe limits since the only involvement of local people was to provide information, while the power of decision-making about the use of this information remained in the hands of researchers and mostly policy makers [56]. With the passing of time, the approach has evolved and refined, given that many actors (mostly international development agencies), operating at grassroots level, improved the techniques in participatory approaches. The concerns and interests of people were now put at the center and the sharing of analysis results, decisions, and planning efforts among the community members were taken into account by researchers and decision-makers. The new paradigm for participatory approaches, put at the center a more demand-responsive ways of managing risk, interaction with nature, and process-oriented thinking. Several practices, both in developed and developing countries saw, over the years, many examples of how to implement local participation and technical tools for reducing disaster risks. The main results of this policy were the building up of people's capacities for analyzing their circumstances, their potential and their problems in order to actively decide on changes and transformations [59].

At the beginning of the 1990s, extended concepts of participatory processes and interaction have been developed, summarized under the name of Participatory and Integrated Development Approaches (PIDA). Those approaches are a mixture of government intervention, local workshops and many other collective activities which are embedded in long-term framework of institutionalized activities. One example is the achievement of the grassroots-level planning integration into local and regional planning approaches. This leads to more sustainable and better coordinated prevention risk processes. In addition to this "vertical" integration, PIDA also tries to enhance horizontal integration—meaning the involvement of different organizations and different groups of stakeholders within a region [58]. As a result of several of these participatory approaches and several researches carried out at field level in many countries, communities seem to respond better when taken into account. Researchers have explored that for example in India, where earthquakes risk reduction at community level is

more effective [60,61]. However, to further reduce risks at community-level, it is necessary to have active participation of the community in creating risk maps. In this paper, we therefore argue that participatory approaches and tools for vulnerability assessment, with focus on participatory mapping techniques, can improve disaster damages and losses. In sum, a real participatory approach starts from recognizing how people try to participate, and from understanding how to facilitate, orientate and strengthen this process, adopting a real bottom-up approach. According to some reports from FAO and other international organizations, participatory approaches that include sharing in decision-making and representation of all the community's groups (women, disabled, elderly, minorities, etc.) within regular consultations with experts, common planning, and transparency of decisions and actions can bring several benefits to vulnerable communities [56,62].

Over the past few years, Mexican experiences on disaster management begun to include the involvement of local communities and people in the risk planning, even though at a very preliminary level. However, these experiences show that developing any kind of risk map, including floods, requires a systematic process at a multilevel framework. On one side, the above mentioned legal framework should clearly address the authority to produce and disseminate flood maps; define roles, responsibilities, and obligations of all involved institutions and other stakeholders; and lay down clearly defined processes [55]. Even though the responsibility of developing and implementing flood management programs are assigned within different Mexican laws, administrative mechanisms, and institutional arrangements with local communities should be implemented. It is important to create mechanisms for enabling partnership among stakeholders in the participatory planning and in the flood mapping program, where available. Establishing community mechanisms, such as a flood mapping committee that represents all the major institutions that have to play an active part in the program and address the interests of all stakeholders, is essential. Therefore, local vulnerable communities are no longer seen as simple recipients; rather, they become critical stakeholders who have a major role to play in the management of community flood map programs.

Integrated Flood Management (IFM) practices based in participatory approaches seek practical ways to maximize the net benefits. Engaging the community throughout the whole project cycle of flood management (assessment, design, implementation, monitoring, and evaluation) at local-level is crucial to create a system where measures undertaken by the community are equitable and effective and where the needs and priorities of the entire affected population are met in the long term. An important step to tackle floods risk can be therefore an integrated approach that brings together different stakeholders and management measures that include regulations, plans, strategies at local-level, community beneficiaries, and coordination/communication mechanisms. Some important steps to a good management approach include the model described in Figure 6.

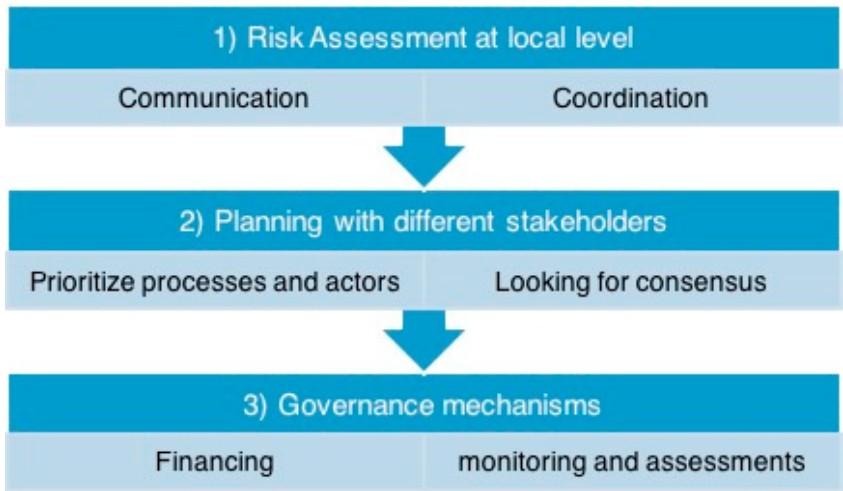

**Figure 6.** Schematic model of a hazard management.

The social model suggests a methodology to combine different approaches to integrate new forms of actions to define risk mapping and in particular flood risk mapping at community level [63]. This is also called a governance model for disaster risk reduction.

- In the case of the Risk assessment at local level (point 1 in Figure 6), the approach must be selected based on flood type and local characteristics. It reflects different stakeholders' specific needs and objectives and includes communication and coordination tools.
- The Planning with stakeholders (point 2 in Figure 6), refers to the role of authorities to reach out consensus between stakeholders. This is a huge problem in rural and vulnerable Mexican communities, where the common land is run by different actors who, in very extraordinary cases, do cooperate for common land management before and during disasters.
- Within the Governance mechanisms (point 3 in Figure 6), we include, wherever possible, measures of regular monitoring and evaluation of implementing measures for flood mapping. Moreover, a regular inspection, maintenance, repair, and replacement work are required for sustainable use of resources. Finally, a crucial step is to involve the financial dimension, which is often the main cause for project failing in risk mapping at local level in Mexico [64].

With an organizational mechanism like the one described here, the use of susceptibility flood maps would be regularized and the education to use them would be an easier goal to reach, at the population as well as governmental decision-maker level. Critical knowledge gaps can also be overcome with the use of such approach. Building disaster resilience at all levels, especially in disaster-prone least developed countries, is an increasing duty for many stakeholders, not only official ones. Spatial information on risk, resources, and capacities of communities must be taken into consideration when planning local disaster prevention policies. However, new efforts are required and general methodological approaches that integrate community-based participatory mapping processes should be carried out in Mexico. As widely demonstrated in other regions of the world, collaborative mapping techniques can improve the way we deal with disasters and emergencies in vulnerable areas [65].

## 6. Conclusions

The change in the frequency of heavy rainfall and a higher vulnerability of watersheds to intense rainfall over the past few years have increased the risk of flooding in various parts of Mexico. Heavy rainfall has become more intense and frequent now than in previous decades, partially in relation to more tropical cyclones. On the other hand, as the dynamics of vulnerability to rainfall is increasing due to continuous land use changes, the risk of floods has risen. The number of floods has grown in areas where physical and social vulnerability have also increased.

In Mexico, the lack of a policy dedicated to the prevention of damage caused by extreme natural phenomena, combined with a decentralization of the organizations responsible for protecting the population, leads to the continuous perpetuation of disasters and damage to the country's economy. The main problem identified here is that while detailed technical methodologies for calculating, modeling, and mapping flood prone areas and flood risks are available in many forms, including data, modeling, and other technical tools, the guidance on overall approaches to flood mapping, and risk assessments at community level are conspicuously missing, particularly where vulnerable population conditions are present. There is particular need for addressing such situations in developing countries like Mexico, where limited know-how, lack of resources, and inadequate data availability as well as difficulties to change risk perceptions in vulnerable areas are common. The situation is even more difficult in transboundary basins due to the absence of acceptable common methodologies for risk assessments and management practices. Moreover, flood risk management and risk perception cannot be treated in isolation—rather it should be a part of community development. It is therefore crucial to propose a community's capacity building to understand local vulnerabilities, strategies, and activities in the form of community planning together with technical tools.

From the analysis carried out it appears that the first step to be taken in Mexico is to improve the organization of the organs predisposed to the civil protection of the country. From a purely technical point of view, we are aware that a probabilistic analysis of the flood phenomenon depends on a complex system of parameters that involve different hydrological and physical processes acting on different time scales. Obtaining a correct parameterization of the physical processes that characterize the phenomenon of flooding is influenced by different types of uncertainties, many of which depend on the error in the measurements in the field. The probabilistic approach and the improvement of research techniques in the field can help to minimize these errors. For example, risk maps, combined with a quantification of the urban drainage system (especially in the case of urban floods) can represent a fundamental tool for floods mitigation and for future adaptation strategies [66]. An important aspect that is beginning to take importance in the forecast of flood phenomena is the identification of nonstationary flooding series and the analysis of their causes [67]. The analysis of the nonstationary flood behavior is an aspect that must be taken into consideration for the development of new risk map construction techniques, for the importance of identifying the variance change points in flood series. All new techniques should be applied in the framework of an integrated approach so to obtain reliable results for both short and long-term flooding forecast. In this sense it is a priority that model developers, meteorologists, hydrologists, social scientists, and stakeholders in general should work together to develop new strategies. A more efficient communication between communities is needed so that the scientific community, when called upon to find easily understandable and usable research solutions, can come into close contact with the bodies prepared for civil protection purposes. This approach to natural risk management is generally applicable, not just to developing countries. As pointed out by Kron [1], nowadays, the vulnerability to flood phenomena has grown enormously all over the world since belongings susceptible to being damaged by water increased. In this context, if forecast is not available, flooding maps are of fundamental importance to increase the preparation for a flood event, both in case of calm and during unrest. A good application of the use of these maps depends enormously on strengthening the collaboration between research and institutions, which must have a univocal view of the problem and must agree on how to operate.

**Author Contributions:** Conceptualization, R.B.; methodology, R.B. and S.L.; formal analysis, R.B. and S.L.; investigation, R.B. and S.L.; resources, R.B. and S.L.; writing—original draft preparation, R.B.; writing—review and editing, R.B. and S.L.; visualization, R.B. and S.L.; supervision, R.B.; project administration, R.B.; funding acquisition, R.B."

**Funding:** This research received no external funding.

**Acknowledgments:** Rosanna Bonasia thanks the Consejo Nacional de Ciencia y Tecnología (CONACYT), Mexico, for the fellowship program Cátedras CONACYT.

**Conflicts of Interest:** The authors declare no conflict of interest.

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
