# Peer review of "Linking Flood Susceptibility Mapping and Governance in Mexico for Flood Mitigation: A Participatory Approach Model"

_atmosphere, doi:10.3390/atmos10080424_

Reviewer 1 Report

This is a good review paper on historical flood damages in Mexico and the relevant intervention and protection policies adopted by the government. A minor revision is required before this paper can be accepted for publication.

1. Authors need to clarify the definition of floods investigated in this study, as there exist several different flood types including fluvial floods, pluvial floods, coastal floods, and flash floods.

2. In line 28 in the introduction, “The recurrence of floods has increased worldwide causing more damage than any other natural phenomenon.”. Are you sure floods are more damaging worldwide than other disasters? Please revise it to make it more accurate or include references to support this statement.

3. Understanding the potential non-stationarity of floods is critical for flood risk management. Adaptation strategies through updating drainage system could also largely regulate flood risks. Discussions on these two important aspects involved in flood risk assessment should be added in the revision. Below are two relevant references that should be included.

Zhou, Q., Leng, G., Su, J., & Ren, Y. (2019). Comparison of urbanization and climate change impacts on urban flood volumes: Importance of urban planning and drainage adaptation. Science of The Total Environment, 658, 24-33.

Liu, S., et al. (2019). Identification of the non-stationarity of floods: Changing patterns, causes, and implications. Water Resources Management, 33(3), 939-953.

Author Response

Response to Reviewer 1 Comments:

1. Authors need to clarify the definition of floods investigated in this study, as there exist several different flood types including fluvial floods, pluvial floods, coastal floods, and flash floods.

Given the great variety of hydrogeological context that Mexico presents, floods occurred in the country are of different kinds, from pluvial to flash floods, without excluding coastal and fluvial ones. In the description of the main flood events that caused damage in the country, provided in Section 2 (Diagnosis of floods in Mexico) it has been specified that we refer to different flood types and analyze the natural causes, as well as the risk management strategies adopted by government structures designed to minimize damage.

2. In line 28 in the introduction, “The recurrence of floods has increased worldwide causing more damage than any other natural phenomenon.”. Are you sure floods are more damaging worldwide than other disasters? Please revise it to make it more accurate or include references to support this statement. 

This statement is due to Kron, W. Flood Risk=Hazard·Values·Vulnerability. International Water Resources Association 2005, Volume 1, pp. 58-68.  The reference has been added in the text. 

3. Understanding the potential non-stationarity of floods is critical for flood risk management. Adaptation strategies through updating drainage system could also largely regulate flood risks. Discussions on these two important aspects involved in flood risk assessment should be added in the revision. Below are two relevant references that should be included.

Zhou, Q., Leng, G., Su, J., & Ren, Y. (2019). Comparison of urbanization and climate change impacts on urban flood volumes: Importance of urban planning and drainage adaptation. Science of The Total Environment, 658, 24-33.

Liu, S., et al. (2019). Identification of the non-stationarity of floods: Changing patterns, causes, and implications. Water Resources Management, 33(3), 939-953.

We agree with the reviewer, above all for the importance that a quantification of the urban drainage system (especially in the case of urban floods) can have when it is incorporated within the methodology for construction of risk maps. Furthermore, we also agree that the analysis of the non-stationary flood behavior is an aspect that must be taken into consideration for the development of new risk map construction techniques, for the importance of identifying the variance change points in flood series. These concepts have been included in the Conclusions, within the discussion on the progress that must be made in the development of methodologies for flood maps construction. Moreover, the suggested bibliographical reference has been added.

Reviewer 2 Report

This paper investigates the link between hazard maps and governance in Mexico.

Please consider the following comments in the first round of revision:

+The references in the body of the paper must be in ordered. Please follow the journal style.

+The contribution in not clear. In the introduction section clear the contribution of the paper in a separate paragraph. after that the organization of the paper must be given according to the objectives and described contribution.

+ the technological advancements regarding the flood hazard mapping have not been given in the paper. Machine learning methods have been widely used for producing the hazard maps of floods. Journal of Water MDPI and  Journal of Atmosphere MDPI include many papers to include in your paper e.g. Mosavi, A., Ozturk, P. and Chau, K.W., 2018. Flood prediction using machine learning models: Literature review. Water, 10(11), p.1536. and Choubin, et al., 2019. An Ensemble prediction of flood susceptibility using multivariate discriminant analysis, classification and regression trees, and support vector machines. Science of the Total Environment, 651, pp.2087-2096.

+elaborate on your methodology and describe it more. 

+the organization of the paper is not efficient, and some parts are too long, please revise accordingly.

+please describe how the results of your research in Mexico can be adapted by other countries

+give a more clearer description by reader on temporal vs spatial flood mapping, and in the earlier part of paper clarify the difference between river floods, and flash-floods

+the paper needs a proofreading done with a scientific writing expert; some issues regarding the using phrases, citation, grammar and presentation exist in the paper. Furthermore, the short paragraphs in the paper make it difficult to read the paper and follow the story. Most of the tile there is no connection between the paragraphs. Giving the websites links in the body of paper is not appropriate.

+accuracy of the existing flood mapping application is not discussed.

Author Response

Response to Reviewer 2 Comments:

1. The references in the body of the paper must be in ordered. Please follow the journal style.

The reference numbering and the order of citations have been corrected following the journal style: references have been numbered in order of appearance in the text (including citations in tables and legends) and listed individually at the end of the manuscript.

2. The contribution in not clear. In the introduction section clear the contribution of the paper in a separate paragraph. after that the organization of the paper must be given according to the objectives and described contribution. 

In order to clarify aims and scopes of the presented review, the Introduction has been modified providing a more precise description of the objectives of the work:

“One of the objectives of this review is to analyze the main natural, political and social causes of the recurrent occurrence of damage caused by flood phenomena in Mexico. This analysis is aimed at justifying the need for projects completely dedicated to prevention, focused on the use of flood maps that should be integrated within a participatory approach model between researchers, development workers, government agents and local populations.”

The organization of the paper is given according to the objectives specified and the methodology described as in the point 4 of this revision.

3. the technological advancements regarding the flood hazard mapping have not been given in the paper. Machine learning methods have been widely used for producing the hazard maps of floods. Journal of Water MDPI and  Journal of Atmosphere MDPI include many papers to include in your paper e.g. Mosavi, A., Ozturk, P. and Chau, K.W., 2018. Flood prediction using machine learning models: Literature review. Water, 10(11), p.1536. and Choubin, et al., 2019. An Ensemble prediction of flood susceptibility using multivariate discriminant analysis, classification and regression trees, and support vector machines. Science of the Total Environment, 651, pp.2087-2096.

In this review we consider that risk map constitute an “a priori” fundamental tool for the prevention of damage caused by floods. Although the main objective of the paper is not to show progress made in the construction of risk maps, but to demonstrate that their application has brought positive results in areas with a high risk of flooding, we agree with the reviewer that a mention about the new techniques is important to leave points for discussion. For this reason, in Section 4 (Flood maps and existing applications), a mention concerning machine learning methods has been added  and the reference to the citation suggested by the reviewer, has been made. 

“It is important to mention that, recently, new techniques have been developed to make risk maps more precise. Data-driven models are gaining much importance along with physical and numerical models. For example, Machine Learning models are data-driven techniques that can be used for prediction purposes, with minimal inputs, on the basis of historical data. This technique has been applied for short and long term prediction and represents a forecasting method with a high level of accuracy [36]. Mosavi, A.; Ozturk, P.; Chau, K. W. Flood prediction using machine learning models: literature review. Water 2018, Volume 10, pp. 1536.” 

4. elaborate on your methodology and describe it more

On the basis of the review aims specified in the Introduction (see point 2), the methodology adopted in the review has been described as follows:

The principal flood events that caused the greatest damage in the country are described. Given the great variety of hydrogeological context that Mexico presents, floods analyzed here are of various type: from pluvial to flash floods, including also the coastal and fluvial ones;

Risk management and civil protection actions currently operating in Mexico are described, and the principal decentralized parts of the government system dedicated to emergency management are analyzed;

Existing flood hazard maps techniques and application in the world are reviewed.

On the basis of data acquired we perform an analysis of the impact of the use of flood maps in Mexico at social and administrative level and propose a hazard management model based on a participatory approach.

5. the organization of the paper is not efficient, and some parts are too long, please revise accordingly

After having clarified the objectives of the work and the methodology used to reach them, the organization of the paper and the paragraphs developed can be more justified. Each paragraph reflects the methodological points listed in the introduction, which, in turn, serve to achieve the objectives of the work.

The text has been revised and modified according to the reviewer’ suggestion so as to eliminate excessively long parts and make it more readable. 

6. please describe how the results of your research in Mexico can be adapted by other countries

In the Conclusions we added the consideration that, in countries like Mexico, where the decentralized political system for managing risks associated with natural phenomena makes damage mitigation more difficult, the development and application of an integrated approach to obtain reliable results for both short and long-term flooding forecast is fundamental. We emphasize that it is a priority that model developers, meteorologists, hydrologists, social scientists and stakeholders in general should work together to develop new strategies, and that a more efficient communication between communities is needed, so that the scientific community, called upon to find easily understandable and usable research solutions, comes into close contact with the bodies prepared for civil protection purposes. This approach to natural risk management is generally applicable, not just to developing countries. Nowadays, the vulnerability to flood phenomena has grown enormously all over the world since belongings susceptible to being damaged by water have increased. Furthermore, the involvement of the populations in risk management plans must be considered a priority, since in many places in the world people ignore or forget to live in areas exposed to flood risk. For these reasons, the approach presented in this paper is applicable to all those places where coexistence with extreme natural phenomena is not yet achieved and a decentralized civil protection system exists.

7. give a more clearer description by reader on temporal vs spatial flood mapping, and in the earlier part of paper clarify the difference between river floods, and flash-floods

Definitions of river, flash and other types of floods have been improved in Section 2 (Diagnosis of floods in Mexico). 

“Mexico is subject to a great variety of flood phenomena, due to the vastness of its territory, and to the variety of hydrogeological contexts that characterize it. Pluvial floods generate as a consequence of different hydrometeorological phenomena occurring within the Republic throughout the year. Because of its geographical position, Mexico is frequently subject to extreme weather phenomena that produce severe convective storms, tropical cyclones and cold fronts. The country is located within a band of low pressure that forms on the regions of masses of warm water in the tropics, known as the Intertropical Convergence Zone, which is characterized by the formation of intense rains that produce floods year after year. Mexico is also affected by river floods, that verify when the water that overflows from rivers, remains on the surface of lands around them. When this happens, the volume of water that drains on the land through the channels increases with the area of the basin. This occurs mainly in the rivers with major length and in those that reach the coastal plains. Coastal floods are also very common, due to the presence of tropical cyclones that generate storm surges. Although less frequent, Mexico has also been affected by floods caused by hydraulic infrastructure failures.

Floods can be the consequence of slow or rapid processes. Some basins are characterized by having a slow hydrological response. This commonly occurs in areas where the slope of the channel is small and, therefore, the capacity of the rivers decreases considerably causing overflows that produce floods in the surrounding areas. In these cases, floods are generated in a relatively long time (several hours or days). On the other hand, sudden and intense rains can cause small currents that can transform, in a matter of minutes, into violent torrents capable of causing great damage. This phenomenon, known as flash flood, happens mainly in urban areas where the presence of buildings prevents the infiltration of water and practically all the precipitated volume becomes runoff. This flood can also occur in basins located near steep mountains where valleys, ravines and alluvial fans exist. Moreover, flash floods can be the consequence of dams breaking”

More clearer indications on the use of the temporal and spatial variations in the construction of flood maps have been provided in Section 4 (Flood maps and existing applications).

“An effective assessment of flood risk involves the analysis of floods in terms of area and time. Analysis of historic records and flood data have been widely used to estimate flood hazards, especially in cases when hydrologic models that use time series flood data to calculate flood depths cannot be applied. The analysis of topographic and geomorphological data variation in areas subject to flooding, provides indications regarding the spatial distribution of the phenomenon that, combined with the temporal variation, represents an important tool for the flood hazard assessment.”

8. the paper needs a proofreading done with a scientific writing expert; some issues regarding the using phrases, citation, grammar and presentation exist in the paper. Furthermore, the short paragraphs in the paper make it difficult to read the paper and follow the story. Most of the tile there is no connection between the paragraphs. Giving the websites links in the body of paper is not appropriate

The manuscript has been thoroughly revised. Errors in editing, punctuation, and citations have been corrected. The editing of the paragraphs has been modified in various points, connecting paragraphs in order to make the reading easier and more fluid. References to the websites in the manuscript have been deleted and moved to the bibliography.

9. accuracy of the existing flood mapping application is not discussed

Considerations about the accuracy of existing flood mapping techniques have been provided in Section 4 (Flood maps and existing applications). We underlined the progress made recently to reduce epistemic and random errors in flood models: greater accuracy in numerical models, statistical analysis of historical data of floods and rains, application of satellite images to reduce errors due to topographical data, dependence of calculated flood levels on DEMs resolution. 

“Methods currently in use for the construction of flood maps have reached acceptable levels of accuracy. Epistemic uncertainties have been reduced improving numerical techniques, while random errors in field measurements have been overcome by introducing detailed statistical studies of historical rainfall and flood data, as well as studies on the effects of DEM properties (vertical and horizontal resolution) on flood models. In many cases, satellite images, when available, drastically reduce the error due to uncertainties of topographic data. It is important to mention that, recently, new techniques have been developed to make risk maps more precise.”

Moreover, in the Conclusions, further discussions on the progress that must be made in the development of methodologies for flood maps construction have been included.

“Obtaining a correct parameterization of the physical processes that characterize the phenomenon of flooding is influenced by different types of uncertainties, many of which depend on the error in the measurements in the field. The probabilistic approach and the improvement of research techniques in the field can help to minimize these errors. For example, risk maps, combined with a quantification of the urban drainage system (especially in the case of urban floods) can represent a fundamental tool for floods mitigation and for future adaptation strategies [63]. An important aspect that is beginning to take importance in the forecast of flood phenomena is the identification of non-stationary flooding series and the analysis of their causes [64]. The analysis of the non-stationary flood behavior is an aspect that must be taken into consideration for the development of new risk map construction techniques, for the importance of identifying the variance change points in flood series.

Zhou, Q.; Leng, G.; Su, J.; Ren, Y. Comparison of urbanization and climate change impacts on urban flood volumes: Importance of urban planning and drainage adaptation. Science of the Total Environment 2019, Volume 658, pp. 24-33.Liu, S.; Huang, S.; Xie, Y.; Wang, H.; Leng, G.; Huang, Q.; Wei, X.; Wang, L. Identification of the non-stationarity of floods: Changing patterns, causes, and implications. Water Resources Management 2019, Volume 33, pp. 939-953.

Round  2

Reviewer 2 Report

+ in line 30, you mention: "The recurrence of floods has increased worldwide causing more damage than any other natural phenomenon [1]. This fact is due to several factors: population growth" these claims are not correct. Firstly flood is generally not the no.1, if it is in some areas, please back it up with suitable citations. secondly, population growth has never been directly linked to increasing flood. However, population growth increase the hazard and devestation. Please be more careful in making claims and please use correct citations for the rest of paper. 

+Paper provides novel idea on using the hazard maps for efficient governance. and the work is of significance due to climate change. the work is also relevant to Atmosphere journal. However, the technologies that are in practiced today to identify the hazard maps (including Mexico) are not presented. Note that the correct term is "flood susceptibility mapping". and what your paper aims to do is to use the "flood susceptibility mapping" for flood mitigation and policy-making. Thus, I suggest, using these two phrases in your title and list of keywords. 

+To successfully linking flood susceptibility mapping" with flood mitigation and policy-making the paper must provide a concrete position and review on "flood susceptibility mapping". Then the paper can progress on linking. Thus, I suggest, add one paragraph on the technological advancements on "flood susceptibility mapping"  considering the IoT, remote sensing, and machine learning prediction methods. Use the insight that the following papers provide for instance: "Choubin, et al. An Ensemble prediction of flood susceptibility using multivariate discriminant analysis, classification and regression trees, and support vector machines. Science of the Total Environment 651 (2019): 2087-2096." and "Choubin, et al. Flash-flood hazard prediction using Ensembles and Bayesian based machine learning models: application of the simulated annealing feature selection methods. Science of the Total Environment 677 (2019): 2389-2399." also the Journal of Atmosphere included a few paper on this realm e.g., "Chau, K.W., 2019. Integration of advanced soft computing techniques in hydrological predictions. Atmosphere." After a review on the technologies used in hazard maps, you can progress on linking it to policy-making, and land use management. Improve your bibliography with including the mentioned above references and more relevant papers. 

Author Response

p.p1 {margin: 0.0px 0.0px 0.0px 0.0px; text-align: justify; font: 12.0px 'Times New Roman'; -webkit-text-stroke: #000000} p.p2 {margin: 0.0px 0.0px 0.0px 0.0px; text-align: justify; font: 12.0px 'Times New Roman'; -webkit-text-stroke: #000000; min-height: 15.0px} p.p3 {margin: 0.0px 0.0px 0.0px 0.0px; text-align: justify; font: 12.0px 'Times New Roman'; color: #ff2d22; -webkit-text-stroke: #ff2d22; min-height: 15.0px} p.p4 {margin: 0.0px 0.0px 0.0px 0.0px; text-align: justify; font: 12.0px 'Times New Roman'; color: #ff2d22; -webkit-text-stroke: #ff2d22} p.p5 {margin: 0.0px 0.0px 0.0px 0.0px; text-align: justify; line-height: 17.0px; font: 12.0px 'Times New Roman'; color: #ff2d22; -webkit-text-stroke: #ff2d22} p.p6 {margin: 0.0px 0.0px 0.0px 0.0px; text-align: justify; line-height: 17.0px; font: 12.0px 'Times New Roman'; color: #ff2d22; -webkit-text-stroke: #ff2d22; min-height: 15.0px} span.s1 {font-kerning: none}

Response to Reviewer 2 Comments:

1. in line 30, you mention: "The recurrence of floods has increased worldwide causing more damage than any other natural phenomenon [1]. This fact is due to several factors: population growth" these claims are not correct. Firstly flood is generally not the no.1, if it is in some areas, please back it up with suitable citations. secondly, population growth has never been directly linked to increasing flood. However, population growth increase the hazard and devestation. Please be more careful in making claims and please use correct citations for the rest of paper.

The statement made in line 30 (“The recurrence of floods has increased worldwide causing more damage than any other natural phenomenon”), is due to Kron, W. Flood Risk=Hazard·Values·Vulnerability. International Water Resources Association 2005, Volume 1, pp. 58-68. However, we agree with the reviewer to acknowledge that the statement we made generalizes a very broad problem that should be investigated in more detail: natural hazards and associated damage. Since this is not the purpose of the paper, the controversial sentence has been modified has follows:

“The recurrence of floods has increased worldwide representing one of the most costliest phenomenon in terms of economic losses”.

We also recognize the inaccuracy in the sentence which confuses the increase in flood frequency  with the increase of the related hazard. The sentence has been modified as follows:

“The increase in the hazard related to river flooding is due to several factors”.

2. Paper provides novel idea on using the hazard maps for efficient governance. and the work is of significance due to climate change. the work is also relevant to Atmosphere journal. However, the technologies that are in practiced today to identify the hazard maps (including Mexico) are not presented. Note that the correct term is "flood susceptibility mapping". and what your paper aims to do is to use the "flood susceptibility mapping" for flood mitigation and policy-making. Thus, I suggest, using these two phrases in your title and list of keywords.

We thank the reviewer for acknowledging the paper’s contribution. 

With regards to the technologies currently in use for the construction of flood maps, we want to specify that in the section dedicated to the use of maps nowadays (4. Flood maps and existing applications), we wanted to emphasize mainly the positive contribution that flood maps have had in the world in reducing the related risk, rather than reviewing the techniques in use. However, some references regarding the methodologies developed in the few works carried out in Mexico on flood maps, were indicated in the same section with the relative bibliographical references. Likewise, a consideration was made about the lack of a unique methodology applied throughout the Republic. 

About the term “flood susceptibility mapping” we agree that it is the most indicated, and it was included in the title and in the keywords. The new title is the following:

“Linking flood susceptibility mapping and governance in Mexico for flood mitigation: a participatory approach model”.

3. To successfully linking flood susceptibility mapping" with flood mitigation and policy-making the paper must provide a concrete position and review on "flood susceptibility mapping". Then the paper can progress on linking. Thus, I suggest, add one paragraph on the technological advancements on "flood susceptibility mapping"  considering the IoT, remote sensing, and machine learning prediction methods. Use the insight that the following papers provide for instance: "Choubin, et al. An Ensemble prediction of flood susceptibility using multivariate discriminant analysis, classification and regression trees, and support vector machines. Science of the Total Environment 651 (2019): 2087-2096." and "Choubin, et al. Flash-flood hazard prediction using Ensembles and Bayesian based machine learning models: application of the simulated annealing feature selection methods. Science of the Total Environment 677 (2019): 2389-2399." also the Journal of Atmosphere included a few paper on this realm e.g., "Chau, K.W., 2019. Integration of advanced soft computing techniques in hydrological predictions. Atmosphere." After a review on the technologies used in hazard maps, you can progress on linking it to policy-making, and land use management. Improve your bibliography with including the mentioned above references and more relevant papers.

In the first round of revisions, a reference to machine learning models and data-driven techniques has already been added in section 4. In this new version, a paragraph has been added on the technological progress in the construction of flooding maps, and the suggested citations have been included.